

# AntiDMPpred: a web service for identifying anti-diabetic peptides

Xue Chen[1], Jian Huang[2] and Bifang He[1]

[1] Medical College, Guizhou University, Guiyang, China
[2] School of Life Science and Technology, University of Electronic Science and Technology of China, Chengdu, China

## ABSTRACT

Diabetes mellitus (DM) is a chronic metabolic disease that has been a major threat to human health globally, causing great economic and social adversities. The oral administration of anti-diabetic peptide drugs has become a novel route for diabetes therapy. Numerous bioactive peptides have demonstrated potential anti-diabetic properties and are promising as alternative treatment measures to prevent and manage diabetes. The computational prediction of anti-diabetic peptides can help promote peptide-based drug discovery in the process of searching newly effective therapeutic peptide agents for diabetes treatment. Here, we resorted to random forest to develop a computational model, named AntiDMPpred, for predicting anti-diabetic peptides. A benchmark dataset with 236 anti-diabetic and 236 non-anti-diabetic peptides was first constructed. Four types of sequence-derived descriptors were used to represent the peptide sequences. We then combined four machine learning methods and six feature scoring methods to select the non-redundant features, which were fed into diverse machine learning classifiers to train the models. Experimental results show that AntiDMPpred reached an accuracy of 77.12% and area under the receiver operating curve (AUCROC) of 0.8193 in the nested five-fold cross-validation, yielding a satisfactory performance and surpassing other classifiers implemented in the study. The web service is freely accessible at http://i.uestc.edu.cn/AntiDMPpred/cgi-bin/AntiDMPpred.pl. We hope AntiDMPpred could improve the discovery of anti-diabetic bioactive peptides.

# INTRODUCTION

Diabetes mellitus (DM) is a chronic metabolic disease which is characterized by upregulated blood glucose levels (*Sajan et al., 2021*). This disorder was ranked as the 7th leading cause of mortality in 2016 and caused 1.6 million deaths worldwide (*WHO, 2018*). It is predicted that roughly one-half of DM patients globally have not been diagnosed, so they are at higher risk of developing serious complications than those receiving treatment (*Saeedi et al., 2019*). Many bioactive peptides have shown potential anti-diabetic activities and are promising to be alternative treatment measures in the management of diabetes (*Acquah et al., 2022*).

Corresponding authors
Jian Huang, hj@uestc.edu.cn
Bifang He, bfhe@gzu.edu.cn

The occurrence of DM leads to long-term damage and dysfunction of numerous tissues, but the most affected is the vascular system (*Aqib et al., 2019*). There is no cure for DM so far. Conventional insulin injections have been used for DM treatment, but this treatment causes multiple complications such as hyperinsulinemia, pain, and discomfort. Glucose-lowering therapy with anti-diabetic drugs can also be employed to manage DM (*Husain et al., 2019*). However, the existing anti-diabetic agents pose negative impacts on patient health (*Aqib et al., 2019*; *Daliri, Oh & Lee, 2017*; *Toroski et al., 2019*). Another problem is that the cost of these drugs is increasingly high and are not affordable to three-quarters of the global low-income population (*Saeedi et al., 2019*). Since the current therapeutic medications have distinct drawbacks, developing new anti-diabetic drugs to prevent, manage, and treat diabetes is urgently needed.

Many peptides have been demonstrated to have impressive anti-diabetic effects. These anti-diabetic peptides modulate blood glucose levels through multiple mechanisms, such as activating the glucagon-like peptide-1 (GLP-1) receptor (*Pratley et al., 2018*) and inhibiting specific enzymes (*Lu et al., 2019*; *Hatanaka et al., 2012*; *Mojica & de Mejia, 2016*; *Mudgil et al., 2019*). The BioDADPep database has been developed for storing and managing anti-diabetic peptides (*Roy & Teron, 2019*). It holds 2,544 entries with 1,143 unique peptides and their relevant detailed information. Out of 1,143 peptides, 1,097 are natural peptides containing only 20 natural amino acids, while 46 are modified peptides. Each entry includes peptide sequence, peptide source (start position–end position), protein function, assay_method/preclinical/clinical and reference, etc. Most of anti-diabetic peptides in BioDADPep were manually collected from published articles, while a small part of anti-diabetic peptides was from other databases (THPdb, ADP3, SATPdb, etc.). BioDADPep is a valuable resource for the design and discovery of novel anti-diabetic peptides. Machine learning based computational models can effectively identify anti-diabetic peptides. However, there is currently no classifier for predicting such peptides.

Here we propose the first bioinformatics tool, called AntiDMPpred. AntiDMPpred is a machine learning-based model to identify anti-diabetic peptides from sequence information. To improve the ease of use, a web server was constructed for AntiDMPpred (http://i.uestc.edu.cn/AntiDMPpred/cgi-bin/AntiDMPpred.pl). We expect that AntiDMPpred could help users find novel anti-diabetic peptides and further advance peptide drug discovery for diabetes treatment.

## MATERIALS AND METHODS

### Overall workflow of AntiDMPpred

The overall framework of AntiDMPpred is shown in Fig. 1. First, a benchmark dataset with 236 anti-diabetic and 236 non-anti-diabetic peptides was established. Second, each peptide was represented by diverse peptide descriptors that were then concatenated into a high-dimensional feature vector. Third, different feature scoring methods were utilized to obtain the optimal features which were then fed into various machine learning classifiers. This step was to select the optimal combination of machine learning and feature

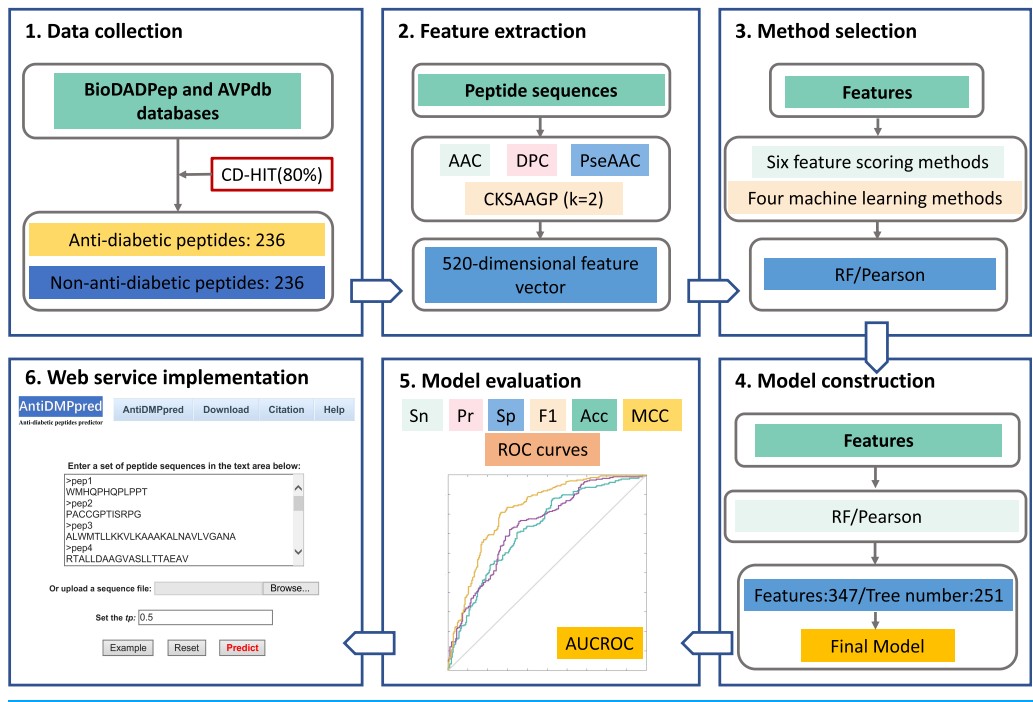

**Figure 1 Workflow of AntiDMPpred.** The computational model was developed through data collection, feature encoding, method selection, model construction and evaluation, and web service implementation.

scoring methods with the highest accuracy in the nested five-fold cross-validation. Finally, the optimal combination was utilized to develop the final model, and the model was then implemented into a web service, which is freely available at http://i.uestc.edu.cn/AntiDMPpred/cgi-bin/AntiDMPpred.pl.

## Datasets

Anti-diabetic peptides were downloaded from the BioDADPep database (*Roy & Teron, 2019*). We collected 2,544 peptide records. Modified peptides were first excluded because the number of such peptides in BioDADPep is very limited. In this study, natural peptides refer to peptides containing only 20 natural amino acids, while modified peptides represent peptides with unnatural residues. After eliminating peptides containing less than five amino acids or more than 50 amino acids, 2,125 anti-diabetic peptides were remained. We then used CD-HIT (*Fu et al., 2012*) to reduce redundant sequences with 80% sequence identity cut-off. Finally, the anti-diabetic dataset consisted of 236 anti-diabetic peptides. For the lack of experimentally validated non-anti-diabetic peptides, the non-anti-diabetic peptides were obtained from the AVPdb database (*Qureshi et al., 2014*). For these peptides, their anti-diabetic activity has not been reported. AVPdb is a database specifically tailored for experimentally validated antiviral peptides (AVPs) that target over 60 viruses and have exhibited tremendous potential in inhibiting viruses (*Qureshi et al., 2014*). It contains 2,683 peptides, of which 624 are modified peptides. We downloaded 2,059 natural peptides where 2,025 peptides had between 5–50 amino acids from AVPdb. After excluding highly similar peptides by CD-HIT (80% sequence identity threshold), the non-anti-diabetic

dataset was composed of 699 peptide sequences. To balance the sample number of anti-diabetic and non-anti-diabetic dataset, we randomly selected 236 peptides from the non-anti-diabetic dataset for training. Finally, the training dataset was comprised of 236 anti-diabetic and 236 non-anti-diabetic peptides.

## Feature extraction

Each peptide was encoded by the following four types of peptide descriptors: amino acid composition (AAC), dipeptide composition (DPC), pseudo amino acid composition (PseAAC) and composition of k-spaced amino acid group pairs (CKSAAGP). These sequence-derived features have been successfully utilized to classify various types of peptides (*He et al., 2016*; *He, Chen & Huang, 2019*; *Li et al., 2017*). Each peptide descriptor was calculated by in-house scripts.

AAC calculates the percentage of each amino acid in a peptide sequence and generates a 20-dimensional feature vector, and it is formulated as:

$$AAC(i) = \frac{x(i)}{\sum_{i=1}^{20} x(i)} \tag{1}$$

where $x(i)$ represents the number of any one of 20 amino acids (*i.e.*, "ACDEFGHIKLMNPQRSTVWY").

DPC, a feature extraction method, calculates the frequency of each amino acid pair within a peptide sequence, generating a 400-dimensional feature vector. It can be defined as follows:

$$DPC(j) = \frac{y(j)}{\sum_{j=1}^{400} y(j)} \tag{2}$$

where $y(j)$ denotes the number of any one of 400 amino acid pairs (*i.e.*, "AA, AC, AD… YY").

The PseAAC compensates for AAC and incorporates sequence-order information by introducing discrete factors. In the calculation formula of PseAAC, the weight factor $\omega$ is designed to put weight to the additional PseAAC with respect to the conventional AAC. $\lambda$ represents the counted rank correlation factor along a peptide sequence. PseAAC generates a vector of $20 + \lambda$ dimensions for each peptide sequence. In this study, we set $\omega = 0.05$ and $\lambda = 5$ to compute PseAAC, so the 25-dimensional feature vector consisted of the first 20 features from AAC and last five features representing the rank-different correlation factors for the sequence-order information. The explicit description of PseAAC can be found at (*Chou, 2009*).

The CKSAAGP feature encoding is transformed from the composition of k-spaced amino acid pairs (CKSAAP) (*Chen et al., 2009*). CKSAAP calculates the frequency of residue pairs separated by k-spaced residues, while CKSAAGP computes the occurrence of group pairs separated by k-spaced residues. For generating CKSAAGP, 20 amino acid residues are first classified into five categories: including the aliphatic residue group (g1: GAVLMI), the aromatic residue group (g2: FYW), the positive charged residue group (g3: KRH), the negative charged residue group (g4: DE), and the uncharged residue group

**Table 1** List of 520 peptide features.

| Peptide descriptor | Feature dimension |
|---|---|
| AAC | 20 |
| DPC | 400 |
| PseAAC | 25 |
| CKSAAGP | 75 |
| (AAC, DPC, PseAAC, CKSAAGP) | 520 |

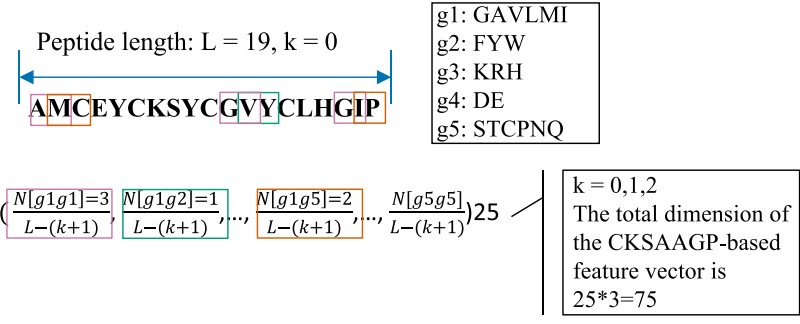

**Figure 2** An illustrated example of the CKSAAGP (k = 0) descriptor.

(g5: STCPNQ) (*Lee et al., 2011*). For a peptide sequence with $L$ amino acids, if $k = 0$, a feature vector of the 25 0-spaced amino acid group pairs (*i.e.*, g1g1, g1g2, g1g3, g1g4, g1g5, …, g5g1, g5g2, g5g3, g5g4, g5g5) is provided in the Fig. 2 and Eq. (3).

$$k = 0, \ \left( \frac{N[g1g1]}{L-(k+1)}, \frac{N[g1g2]}{L-(k+1)}, \frac{N[g1g3]}{L-(k+1)}, \ \cdots, \frac{N[g5g5]}{L-(k+1)} \right) 25 \tag{3}$$

In this work, $k = 0$, 1 and 2 were jointly considered. Thus, the CKSAAGP encoding of each anti-diabetic/non-anti-diabetic peptide has a dimensionality of $25 \times 3 = 75$.

Finally, each peptide was encoded by the four types of feature extraction methods, implemented by the Matlab programming language. After direct mergence of all features, the feature vector with a dimension of 520 was finally obtained (see in Table 1).

## Feature scoring and selection

The 520 features generating by four types of peptide descriptors may contain redundant information or information irrelevant to the classification of anti-diabetic/non-anti-diabetic peptides (*Chowdhury et al., 2020*). These redundant and irrelevant information may lead to a decrease in the predictive performance of the constructed model (*Liang et al., 2021*). Therefore, we adopted six feature scoring approaches, including F-score (Fscore), K-means (Kmeans), Lasso, Pearson correlation (Pearson), Spearman correlation (Spearman) and Student's t-test (Ttest), to rank features in the feature vector (AAC, DPC, PseAAC, CKSAAGP), which was generated by joining the individual feature vector.

We then combined machine learning methods used in the training process (see in the next section) with these feature scoring strategies to select features for model construction based on the accuracy of the five-fold cross-validation.

## Machine learning method and model training

Machine learning classification algorithms, including support vector machine (SVM), random forest (RF) and K-nearest neighbor (KNN), were employed to construct the prediction models to distinguish the anti-diabetic peptides from non-anti-diabetic peptides. For the SVM algorithm, two commonly used kernel functions, linear function and radial basis function (rbf), were employed to implement classification. In this study, we named the SVM with linear function as LinearSVM and SVM with rbf as rbfSVM. Therefore, there are 24 combinations of feature scoring and machine learning methods in this work. LIBSVM (http://www.csie.ntu.edu.tw/~cjlin/libsvm/) was utilized to accomplish SVM (*Chang & Lin, 2011*). RF and KNN were implemented by MATLAB built-in functions. Parameters were optimized based on each subdataset for training in the five-fold cross-validation through a grid search for each model. The parameter combinations that need to be optimized are listed in the Tables S1 and S2. The optimal features and parameters were then fed into different classifiers for model construction that was performed by in-house scripts.

## Model evaluation

The classification models were assessed by a nested five-fold cross-validation that has an inner loop for optimizing models/parameters and an outer loop for evaluating the performance of the models constructed in the inner layer. In this study, in the outer layer of the five-fold cross-validation, the training dataset (236 anti-diabetic peptides and 236 non-anti-diabetic peptides) was randomly split into five equally sized subsets where a subset served as the testing-set and the remaining four subsets as the training-set. In the inner loop, the training-set constructed in the outer layer were equally redivided into five subsets, among which four subsets were utilized for tuning parameters and one for estimating models. The pseudo code for the nested five-fold cross-validation can be found in the Supplemental File nested_CV.txt. We utilized Sensitivity (Sn), Precision (Pr), Specificity (Sp), F1 score (F1), Accuracy (Acc) and Matthew's correlation coefficient (MCC) to evaluate the predictive performance of the models implemented in the study. These metrics are calculated by the following equations:

$$Sn = \frac{TP}{TP + FN} \tag{4}$$

$$Pr = \frac{TP}{TP + FP} \tag{5}$$

$$Sp = \frac{TN}{TN + FP} \tag{6}$$

$$F1 = \frac{2 \times TP \times TP}{TP \times (TP + FN) + TP \times (TP + FP)} \tag{7}$$

$$Acc = \frac{TP + TN}{TP + FN + TN + FP} \tag{8}$$

$$MCC = \frac{TP \times TN - FP \times FN}{\sqrt{(TP + FP) \times (TP + FN) \times (TN + FP) \times (TP + FN)}} \tag{9}$$

where TP denotes the number of true positives, TN as the number of true negative, FP as the number of false positives and FN as the number of false negatives.

The receiver operating curve (ROC) was also plotted and the area under the ROC curve (AUCROC) was calculated to assess the classification results.

## Data augmentation

Data augmentation technology may be potentially more efficient to solve issues of insufficient data, or unbalanced data when data collection is difficult (*Han, Xie & Lin, 2020*; *Mahmud et al., 2021*). This is common to the field of computer vision (*Chaitanya et al., 2021*; *Wang et al., 2021a*). In the field of bioinformatics, data augmentation technology has also been reported to improve the predictive accuracy of models, such as ACP-DA, a predictor for anticancer peptides (*Chen et al., 2021*). In this work, we also used the data augmentation technique to build prediction models to test whether it could improve the prediction performance of classifiers. Noise adding oversampling (NAO) (*Chen, Liu & Zhang, 2022a*) was utilized to augment the positive and negative samples in the feature space, respectively, generating new samples in the feature space. The new samples are generated by calculating as follows:

$$F_{new} = F_i * V * c + F_i \tag{10}$$

where $F_i$ represents a random sample from the training dataset (benchmark dataset), $V$ represents a randomly generated group of 520-dimensional random numbers between 0 and 1 that obey uniform distribution, and $c$ represents the coefficient of the perturbation which was set as 0.02 in this study.

## Final model construction

The final model was constructed by using the RF algorithm based on the training dataset, which was comprised of 236 anti-diabetic peptides and 236 non-anti-diabetic peptides. Feature selection was performed by RF and Pearson correlation. The optimal features were
**Table 2 Performance of the predictions under the combinations of RF with six feature scoring methods.**

| Machine learning method | Feature scoring method | Accuracy (%) | Sensitivity (%) | Specificity (%) | Precision (%) | MCC | F1 | AUCROC |
|---|---|---|---|---|---|---|---|---|
| Random forest | Fscore | 74.79 | 76.27 | 73.31 | 74.07 | 0.4960 | 0.7516 | **0.8202** |
| | Kmeans | 72.88 | 73.31 | 72.46 | 72.69 | 0.4576 | 0.7300 | 0.7933 |
| | Lasso | 73.09 | 76.69 | 69.49 | 71.54 | 0.4631 | 0.7403 | 0.8087 |
| | Pearson | **77.12** | **80.51** | **73.73** | **75.40** | **0.5436** | **0.7787** | 0.8193 |
| | Spearman | 76.27 | 79.66 | 72.88 | 74.60 | 0.5266 | 0.7705 | 0.8163 |
| | Ttest | 74.15 | 78.39 | 69.92 | 72.27 | 0.4848 | 0.7520 | 0.8046 |

**Note**:
The highest metric is highlighted in bold.

then fed into the RF classifier. Selection of the optimal parameter combination was executed through a grid search. All these steps were performed by in-house scripts. The pseudo code for final model construction can be found in the Supplemental File final-model-construction.txt. The model construction and evaluation were implemented at a computational server (Sugon I840-G20: Dawning Information Industry Co., LTD., Beijing, China).

# RESULTS

## Comparison with other machine learning methods

To select the optimal pair of feature scoring and machine learning methods, we combined six feature scoring approaches with four machine learning algorithms in this study. The performance of these 24 combinations in the five-fold cross-validation is given in the Supplemental File nested_CV_results.xlsx. Five-fold cross-validation results of the combinations of RF with six feature ranking strategies are shown in Table 2. The pair of RF and Pearson achieved an accuracy of 77.12% with 80.51% sensitivity, 73.73% specificity, 75.40% precision, 0.5436 MCC, 0.7787 F1 and 0.8193 AUCROC, which suggests that the selected features are able to capture the different discrepancies between anti-diabetic and non-anti-diabetic peptides. Compared with other combinations, the RF/Pearson combination outperformed the other five combinations for all evaluation indicators except AUCROC. The AUCROC achieved the highest level under the combination of Fscore and RF, and AUCROC scores under all combinations are not much different. The ROC curves of the above six predictors are presented in Fig. 3A.

The performances of the classifiers constructed by the combinations of Pearson and four machine learning algorithms are shown in Table 3. It is observed that the RF-based predictor outperformed the other three machine learning predictors, and it's all seven evaluation metrics reached the highest level. The ROC curves of these predictions are illustrated in Fig. 3B. Note that the ROC curve of the KNN-based model cannot be drawn, and its AUCROC value is unable to be calculated.

## Compared with data augmentation method

Here, we used data augmentation technology to randomly sample the original positive and negative samples 100, 200 and 300 times respectively, and finally obtained three new

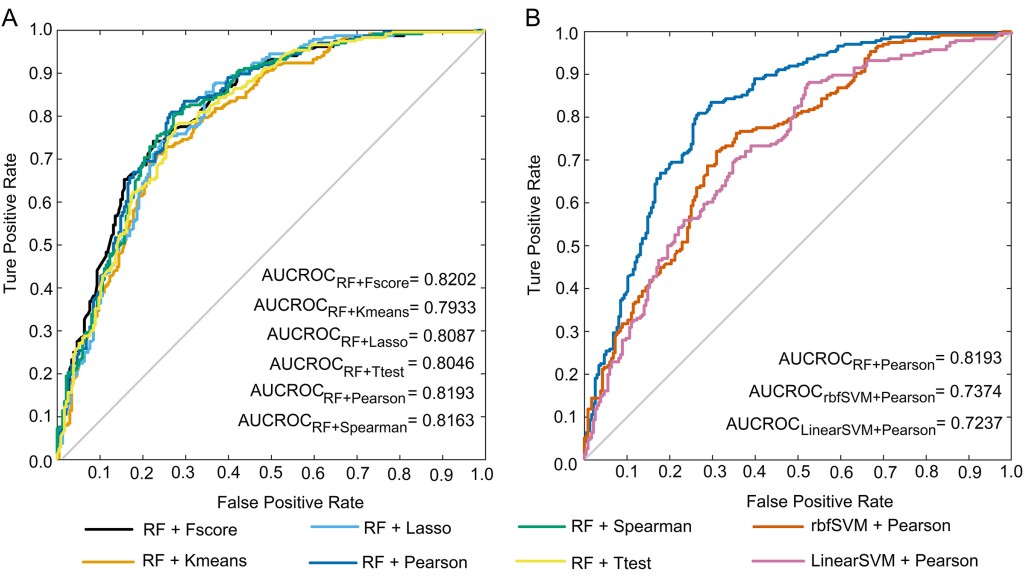

**Figure 3 ROC curves.** (A) ROC curves of the predictions under the combinations of RF with six feature selection methods. (B) ROC curves of the predictions under the combinations of Pearson with three machine learning methods.

**Table 3 Performance of the predictions under the combinations of Pearson with four machine learning methods.**

| Feature selection | Machine learning method | Accuracy (%) | Sensitivity (%) | Specificity (%) | Precision (%) | MCC | F1 | AUCROC |
|---|---|---|---|---|---|---|---|---|
| Pearson | KNN | 72.46 | 76.69 | 68.22 | 70.70 | 0.4508 | 0.7358 | / |
| | LinearSVM | 67.58 | 70.76 | 64.41 | 66.53 | 0.3524 | 0.6858 | 0.7237 |
| | RF | **77.12** | **80.51** | **73.73** | **75.40** | **0.5436** | **0.7787** | **0.8193** |
| | rbfSVM | 69.07 | 68.64 | 69.49 | 69.23 | 0.3814 | 0.6894 | 0.7374 |

**Note**:
The highest metric is highlighted in bold.

datasets to build predictive models, which were namely as 100% dataset, 200% dataset and 300% dataset respectively. Nested five-fold cross-validation results of the combinations of RF with Pearson strategy for different datasets are shown in Fig. 4. Generally, compared with the results of the original dataset, other model prediction performance metrics of the data augmentation technique on the RF algorithm are not improved, but decreased, except for AUCROC and Pr values when using 300% dataset to construct model. For other three machine learning algorithms, there is also no improvement of predictive performance (see Table S3). Therefore, the data augmentation technology was not applied to the construction of our final model.

### The construction and performance of AntiDMPpred

The AntiDMPpred classifier was implemented by using the RF with 347 features that were selected by Pearson. The optimal number of decision trees in this model is 251. As depicted in Table 3, AntiDMPpred achieved an accuracy of 77.12% and AUCROC of 0.8193 with 80.51% sensitivity, 73.73% specificity, 75.40% precision, 0.5436 MCC and 0.7787 F1 in the nested five-fold cross-validation.
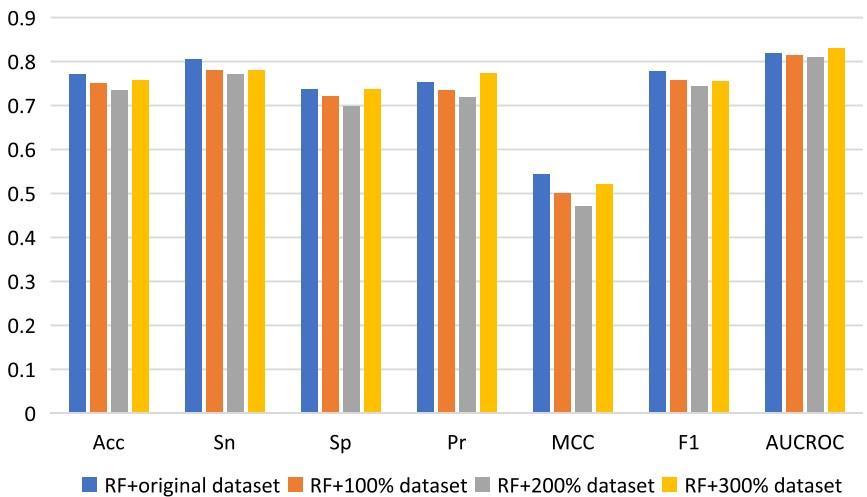

**Figure 4 Performance assessment for data augmentation under the combination of RF with Pearson.**

## Assessment of AntiDMPpred by using other peptide databases

To further evaluate the performance of the AntiDMPpred predictor, 42,973 peptides were collected for testing. We conducted the following procedures to obtain these peptides: (1) collecting 68,390 peptides from different types of peptide databases, including BioPepDB (*Li et al., 2018*), CAMPR3 (*Waghu et al., 2016*), CancerPPD (*Tyagi et al., 2015*), CPPsite (*Gautam et al., 2012*), DRAMP 2.0 (*Kang et al., 2019*), HIPdb (*Qureshi, Thakur & Kumar, 2013*), SATPdb (*Singh et al., 2016*), THPdb (*Usmani et al., 2017*) and UniProt with query: "(type:peptide) length:(5 TO 50) AND reviewed: yes" as candidate testing data; (2) preserving the unique peptide sequences containing only 20 natural amino acids; (3) removing the sequences of less than five amino acids or more than 50 amino acids; (4) eliminating the amino acid sequences that appeared in the BioDADpep and AVPdb. It should be noted that whether these peptides have anti-diabetic properties has not been reported. Table 4 demonstrates the detailed information and predictive results of these peptides. Average 75.02% of the testing data are predicted to be non-anti-diabetic peptides, suggesting that our predictor is capable of effectively learning useful feature representations for distinguishing anti-diabetic peptides from non-anti-diabetic peptides.

## AntiDMPpred web service

To better serve the scientific community, a user-friendly web service was constructed for AntiDMPpred, which is freely available at http://i.uestc.edu.cn/AntiDMPpred/cgi-bin/AntiDMPpred.pl. The server-side processes were performed by Matlab scripts, and Perl and HTML were used to develop the web interfaces. AntiDMPpred allows users to input the query peptide sequences into the input box or upload a sequence file (Fig. 5A). Eventually, the predictive results can be obtained after pressing the Predict button (Fig. 5B).

Users can also modulate the threshold of the probability (*tp*) whose value can range from 0 to 1 to differentiate between predicted positives and negatives. In this work, the

**Table 4 Detailed descriptions and predicted results of other databases.**

| Database | Total number of peptides | Number of unique peptides for testing | Sp (%) |
|---|---|---|---|
| BioPepDB | 4,807 | 3,475 | 74.47 |
| CAMPR3 | 8,225 | 3,694 | 71.95 |
| CancerPPD | 3,142 | 404 | 91.83 |
| CPPsite | 1,564 | 1,125 | 92.80 |
| DRAMP 2.0 | 28,023 | 20,824 | 83.63 |
| HIPdb | 9,81 | 839 | 53.52 |
| SATPdb | 16,590 | 9,410 | 77.56 |
| THPdb | 894 | 17 | 58.82 |
| UniProt | 4,164 | 3,185 | 70.61 |
| Average | | | 75.02 |

Note:
Sp represents the percentage of peptide sequences that are predicted to be non-anti-diabetic peptides.

**Table 5 Performance of AntiDMPpred in the five-fold cross-validation when tp changes.**

| tp | Accuracy (%) | Specificity (%) | Sensitivity (%) | Precision (%) | MCC | F1 |
|---|---|---|---|---|---|---|
| 0.05 | 50.21 | 0.4237 | 100.00 | 50.11 | 0.0461 | 0.6676 |
| 0.1 | 50.85 | 1.69 | 100.00 | 50.43 | 0.095 | 0.6705 |
| 0.2 | 56.57 | 13.56 | 99.58 | 53.53 | 0.2576 | 0.6963 |
| 0.3 | 64.41 | 30.51 | 98.31 | 58.59 | 0.3920 | 0.7342 |
| 0.4 | 70.76 | 48.73 | 92.80 | 64.41 | 0.4626 | 0.7604 |
| 0.5 | 77.12 | 73.73 | 80.51 | 75.40 | 0.5436 | 0.7787 |
| 0.6 | 67.59 | 86.86 | 48.31 | 78.62 | 0.3812 | 0.5984 |
| 0.7 | 58.48 | 97.03 | 19.92 | 87.04 | 0.2662 | 0.3241 |
| 0.8 | 52.12 | 99.58 | 4.66 | 91.67 | 0.1346 | 0.0887 |
| 0.9 | 50.00 | 100 | 0 | / | / | 0 |
| 0.95 | 50.00 | 100 | 0 | / | / | 0 |

assessment of model performance under different $tp$ values can serve as a reference for $tp$ value selection. Detailed results are shown in Table 5, with the increase of the $tp$ value, the value of sensitivity decreases, while the specificity increases. When $tp$ is 0.5, the value of specificity and sensitivity is 73.73% and 80.51%, and ACC, MCC and F1 achieves the highest score of 77.12%, 0.5436 and 0.7787, respectively. This is the reason why the $tp$ value was set as 0.5 by default. When $tp$ is 0.95, the value of specificity is 100%, while the value of sensitivity is 0.

## DISCUSSION

Compared with the current costly experimental verification, there has been growing interest in using machine learning (ML) methods for peptide activity prediction (*Basith et al., 2020*). The construction of anti-diabetic peptide predictors may promote the discovery of anti-diabetic peptides and even boost the development of anti-diabetic drugs.

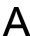

A

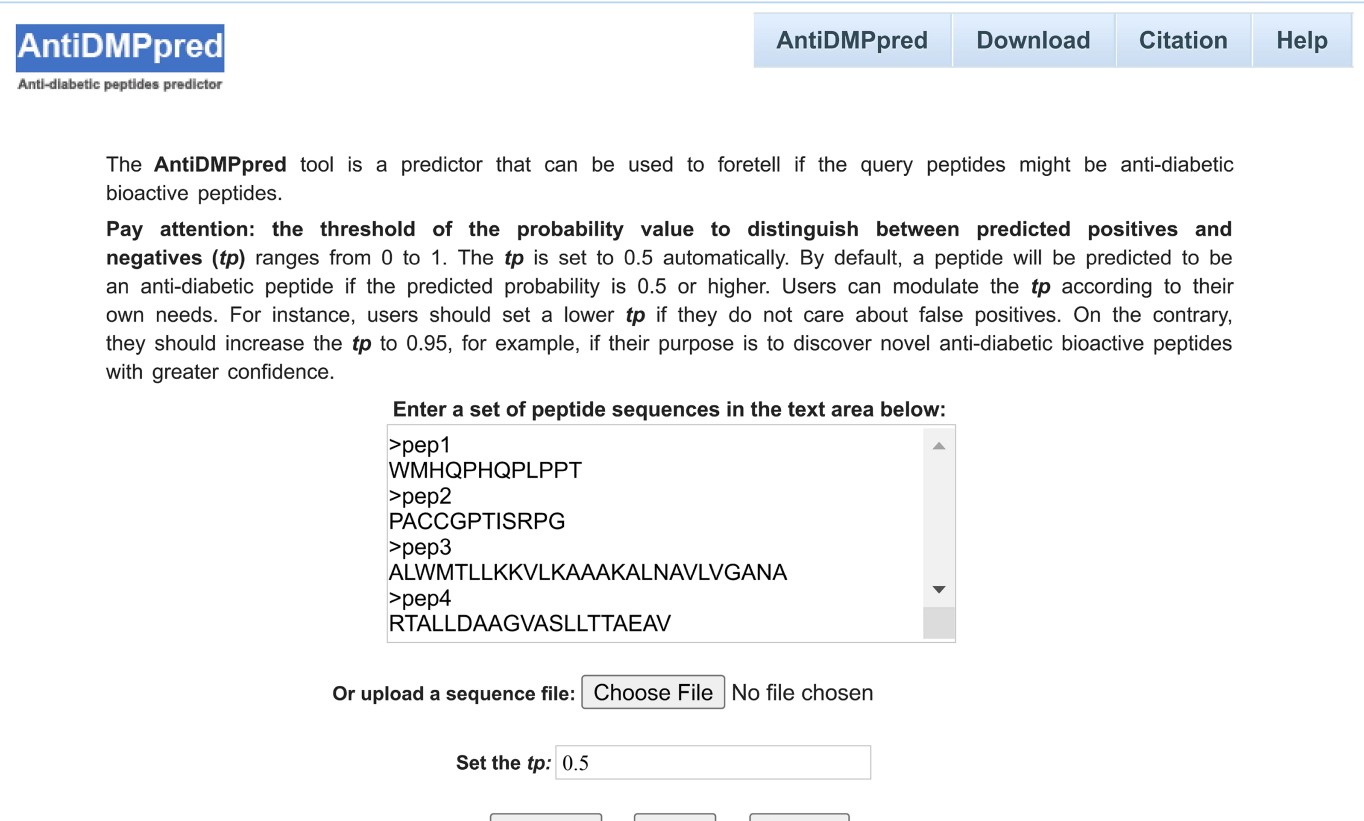

B

All predictive results are displayed in the following table. You can click ,**Number**, **Query Sequence**, **Length** or **Probability** to sort the results in ascending or descending order. The **"Probability"** column shows the probability value that the query sequence is predicted to be an anti-diabetic peptide, which is obtained by the probability value of the RF-based model. When the **"Yes/No"** column is "Yes", it indicates that the sequence is predicted to be an anti-diabetic peptide.

| Number ⇕ | Query Sequence ⇕ | Length ⇕ | Probability ⇕ | Yes/No |
|---|---|---|---|---|
| 1 | WMHQPHQPLPPT | 12 | 0.75 | Yes |
| 2 | PACCGPTISRPG | 12 | 0.83 | Yes |
| 3 | ALWMTLLKKVLKAAA KALNAVLVGANA | 27 | 0.18 | No |
| 4 | RTALLDAAGVASLLTT AEAV | 20 | 0.92 | Yes |

**Figure 5** **Web interface of AntiDMPpred.** (A) The query sequences and threshold of the probability value (tp) are required to be submitted in the input interface. (B) Predictive result interface displaying the query sequences, sequence length, predicted probability and prediction class.

However, an anti-diabetic peptide predictor is currently lacking, and it is crucial to learn from other peptide predictors. Anticancer peptide predictors based on ML methods have been a research hotspot in the field of bioinformatics in recent years, and more than a dozen of anticancer peptide predictors have been successfully developed, such as ACP-Fuse (*Rao et al., 2020*), AntiCP 2.0 (*Agrawal et al., 2021*), CpACpP (*Nasiri et al., 2021*) and CL-ACP (*Wang et al., 2021b*). Most of them were designed based on conventional ML algorithms, such as SVM, RF and Naïve Bayes (NB) (*Wang et al., 2021b*), and SVM is the most popularly used (*Liang et al., 2021*). Partial predictors were achieved based on deep learning (DL) methods, such as long short-term memory neural network, and convolutional neural network. Generally, both traditional ML methods and DL methods use various features, such as AAC, DPC, PseAAC, Composition/Transition/Distribution (CTD) and Grouped Di-Peptide Composition (GDPC), as input to train prediction models. The successful implementation of these anticancer peptide predictors based on ML methods has greatly promoted the discovery of new anticancer peptides. ML methods were also applied in other peptide predictors, such as antimicrobial peptides (*Xiao et al., 2021*), blood-brain barrier penetrating peptides (*Chen et al., 2022b*), streptavidin-binding peptides (*He et al., 2016*) and therapeutic peptides (*Wei et al., 2019*). In our work, we also used the RF algorithm to achieve the first anti-diabetic peptide predictor, which may play a role in the discovery of new anti-diabetic peptides.

The construction of a high-quality benchmark dataset is the first step for the successful implementation of the prediction model (*Charoenkwan et al., 2020a*). In the present study, the benchmark dataset consisted of 236 anti-diabetic peptides from BioDADPep and 236 non-anti-diabetic peptides from AVPdb. However, the non-anti-diabetic peptides have not been experimentally identified with no anti-diabetic activity. We compared the data in BioDADPep database with many other existing peptide databases and analyzed the possibility of AVPdb database containing anti-diabetic peptides. We downloaded peptide sequences from AVPdb, BioPepDB, CancerPPD, CAMPR3, CPPsite, DRAMP 2.0, HIPdb, SATPdb, THPdb and UniProt with query: "(type: peptide) length: (5 TO 50) AND reviewed: yes". After excluding peptide sequences with unnatural residues ("X", "B" and "Z", etc.) and eliminating peptides containing less than five amino acids or more than 50 amino acids, the remaining peptide sequences in each database were compared with anti-diabetic peptides from BioDADPep. It was found that only BioPepDB and SATPdb databases had overlapping peptides with the BioDADpep database, and the number was only 19 and 20, with a frequency of 0.51% and 0.13%, respectively. The average frequency of anti-diabetic peptides in the 10 peptide databases is very small with only 0.068%. Detailed descriptions of these databases are listed in Table S4. Moreover, the BioPepDB database is a database of food-derived bioactive peptides, and part of anti-diabetic peptides are derived from milk protein as the same as some peptides of BioPepDB. SATPdb is a database of structurally annotated therapeutic peptides, which is the combination of many peptide databases. It is not surprising that these two databases have overlapping peptides with the BioDADpep database. Therefore, we can speculate that the probability of anti-diabetic peptides appearing in other peptide databases is very low and we consider the peptides in AVPdb as non-anti-diabetic peptides. In the future, if the data of non-anti-

diabetic peptides verified by experiments are enough to construct a predictor, we will further improve the anti-diabetic peptide predictor using experimentally verified non-anti-diabetic peptides.

## LIMITATION

In the present study, we constructed the negative training dataset by collecting peptides from the other database (AVPdb in this study), which is a common-used strategy for peptide predictor studies (*Ma et al., 2022*; *Chu et al., 2022*; *Charoenkwan et al., 2020b*). Although it is of low probability that these peptides are anti-diabetic peptides, they might show anti-diabetic activity. We will update the AntiDMPpred predictor when sufficient experimentally validated non-anti-diabetic peptides emerge.

## CONCLUSIONS

The accurate and fast identification of anti-diabetic bioactive peptides can help speed up peptide-based drug discovery in the process of searching newly effective therapeutic peptide agents for diabetes treatment. We have proposed AntiDMPpred, the first web service to predict anti-diabetic peptides from sequence information based on RF and hybrid features. We demonstrated its performance through a nested five-fold cross-validation. The results show that the AntiDMPpred model has achieved promising results. Its AUCROC value reached 0.8193, and its MCC value reached 0.5436. It is freely accessible at http://i.uestc.edu.cn/AntiDMPpred/cgi-bin/AntiDMPpred.pl.

### Funding

This work was supported by the National Natural Science Foundation of China (Grant Numbers: 61901130, 61901129, and 62071099), Science and Technology Department of Guizhou Province (Grant Numbers: [2020]1Y407, ZK[2022]-General-056 and ZK [2022]-General-038) and Guizhou University (Grant Numbers: (2018)54, (2018)55 and (2020)5). The funders had no role in study design, data collection and analysis, decision to publish, or preparation of the manuscript.

### Grant Disclosures

The following grant information was disclosed by the authors:
National Natural Science Foundation of China Grant Numbers: 61901130, 61901129, and 62071099.
Science and Technology Department of Guizhou Province Grant Numbers: [2020]1Y407, ZK[2022]-General-056 and ZK[2022]-General-038.
Guizhou University Grant Numbers: (2018)54, (2018)55 and (2020)5.

### Competing Interests

The authors declare that they have no competing interests.

![PeerJ]

## Author Contributions

- Xue Chen performed the experiments, analyzed the data, prepared figures and/or tables, authored or reviewed drafts of the article, and approved the final draft.
- Jian Huang conceived and designed the experiments, authored or reviewed drafts of the article, and approved the final draft.
- Bifang He conceived and designed the experiments, prepared figures and/or tables, authored or reviewed drafts of the article, and approved the final draft.

## Data Availability

The dataset is available at AntiDMPpred: http://i.uestc.edu.cn/AntiDMPpred/download.html.

## Supplemental Information

Supplemental information for this article can be found online at http://dx.doi.org/10.7717/peerj.13581#supplemental-information.

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
