# Peer review of "AntiDMPpred: a web service for identifying anti-diabetic peptides"

_PeerJ, doi:10.7717/peerj.13581_

## Round 0.1 · original submission · Major Revisions

You have two reviews to consider, both of which were enthusiastic and both of which recognise the value of the work. One is from a computational biologist, the other from a peptide expert. Their comments are quite extensive (hence, I have chosen 'major revision') but my overall impression is that most of these are relatively straightforward to address and reflect attempts to improve the clarity and accessibility of your work.

There are also some helpful comments about the on-line tool.

The two comments that require potentially significant changes are from reviewer-1 - how was it ensured that the "control" peptides had no anti-diabetic activity, and is it possible to consider how the stated accuracy statistics change as the threshold changes. These points require careful consideration, but all issues raised will need to be addressed.

Please provide a detailed point-by-point discussion of these comments when you resubmit.

·

Excellent Review

This review has been rated excellent by staff (in the top 15% of reviews)
EDITOR COMMENT
A detailed and cogent review from a reviewer willing to be made public, reflecting the balanced and fair ideas it contains. Helpful and cogent with clear statements and no ambiguity.

Basic reporting

The introduction section provided a good background to antidiabetic peptide therapeutics, setting the scene for the application of machine learning approaches and the objectives of the study. A little bit more background on BioDADPep would have been useful, particularly as this resource was used as the exclusive source of antidiabetic peptides in the data collection.

However, the paper is lacking a view of comparable tools or services. The authors state that there are no other anti-diabetes peptide predictors, and that is true, but it should be possible to appraise wider developments that are relevant e.g. taking a glance across at anticancer peptide predictors to see what kind of methods they use and what kind of benchmarking scores they're putting out. It's not a like for like comparison but would yield valuable reflection and context.

This is particularly important as the abstract states "surpassing other state-of-the-art classifiers" but there is no evidence presented for that claim, indeed as made clear in the introduction "there is currently no classifier for predicting such peptides". There is perhaps a contradiction here, unless what the authors mean by "other state-of-the-art" is their own SVMs and kNN that they implemented in the paper (which performed less well than Random Forest), which in that case as currently written is misleading and should be clarified.

Figure 1 should be associated with the methods section rather than the introduction.

The cited papers were well referenced and relevant. Ensure consistency with no use or use of hyphen in "antidiabetic".

Good use of figures and the legends are clear. I'm not sure about the visual resolution. The tables are fine.

Raw data fine.

Experimental design

The study represents original primary research within the scope of the journal. The important research question was very well defined, as is the targeted knowledge gap. The methods were outlined clearly.

Methods - with the AVPdb database, how was it ensured that the "control" non-antidiabetic peptides had no antidiabetic activity in reality? If this detail is contained in the annotation of AVPdb, then this should be explained. If, however, information from the AVPdb can not be used to confirm whether peptides are in reality devoid of any antidiabetic property, this is an aspect that should be considered in the design of the study. Otherwise, the study could be inherently incorporating false negatives inappropriately in the training sets. "Negative" sets are sometimes really "unknown" sets and whether this is the case should be clarified here - and either way, how well that maps to the desired use of the tool and its interpretation.

Validity of the findings

The findings appear valid, though the question relating to ensuring that the "control" non-antidiabetic peptides had no antidiabetic activity in reality needs to be addressed.

The study resulted in the development of a useful tool. The reported accuracy of 77.12% and area under the receiver operating curve (AUCROC) of 0.8193 in the nested five-fold cross-validation is good, but by no means exceptional, for ML classifiers and might be further improved. Data augmentation techniques, such as those used in this paper - ACP-DA: Improving the Prediction of Anticancer Peptides Using Data Augmentation https://www.frontiersin.org/articles/10.3389/fgene.2021.698477/full - might be used to improve the scores somewhat.

Though there is no reason to doubt the stated statistics for performance for the "known antidiabetes vs known not" sets, when we tested a CYP2D6 sequence chopped up into 30-aa "peptides", the following predictions were generated (please see uploaded image) - 13 peptides were predicted to be antidiabetic peptides, 4 not. This is a very high level of positive prediction for what is "nonsense" data in this regard.

This is probably due to the Random Forest classifier not having been trained on data like this, and so it is generating a high proportion of "low confidence" predictions (e.g. near 0.5), marginal "yes" decisions, though some scores are markedly higher.

I note that the website suggests applying a modified threshold of "0.95" to "discover novel peptides with greater confidence", with a box allowing manual adjustment of the threshold. This is also mentioned in passing in the paper itself and appears to be a very important feature to increase the confidence we can have about the positive predictions.

This brings us to the central point regarding stated accuracy - is it possible to produce analysis and discussion about how the model's various accuracy statistics change, as the threshold changes? e.g. What's the accuracy, sensitivity, and precision of the approach when a user chooses 0.90, 0.95, 0.99? This analysis should be accompanied by a discussion of the envisioned use-cases for a 0.5 threshold vs. a 0.95 threshold, for example.

This would be valuable as it would allow users to know what level of greater (real-world) confidence in positive predictions they can actually have when following the recommendation to use the system at e.g. "0.95" vs the default "0.5".

This would give the user a way to attain some degree of certainty (e.g. t=0.95 might mean lots of false negatives but a very high PPV, so if you ever get a "yes" you can be sure it's a strong "yes" and is worth investigating further). This area needs further elaboration.

The AntiDMPpred web service itself is a tidy web development that appears to work effectively. It's fast with a nice straightforward interface.

There were no papers cited in the results and discussion section. The findings should be linked back to the relevant literature, which as alluded to above, needs to be expanded upon.

The conclusions are generally straightforward and measured. However, there is no way to corroborate the statement "AntiDMPred......will play a critical role in improving the development of useful therapeutic antidiabetic agents". It could indeed play a pivotal role because it is the first machine learning that has successfully targeted this research question, which is commendable enough in itself.

·

Excellent Review

This review has been rated excellent by staff (in the top 15% of reviews)
EDITOR COMMENT
A constructive and detailed review which provides detailed and clear feedback to the authors, with the aim of improving the 'readability' of the paper. This impressed for two reasons: a clear statement of the knowledge of the reviewer enables the authors to understand where their paper may fall-short of clarity to the non-expert bioinformatician, and then the reviewer added suggestions for the web tool too - extra benefit!

Basic reporting

The manuscript, AntiDMPpred: a web service for identifying anti-diabetic peptides, by X. Chen, B. He, and J. Huang, reports a new web-based tool that predicts anti-diabetic peptides. This tool has the potential to aid researchers in identifying lead peptides for further studies of bioactive molecules aimed at the treatment of diabetes.

The manuscript is assembled in a professional manner, with sections appropriate for the content of the study. It is self-contained, and the amount of content is appropriate for one publication unit. It is organized, generally well-written, and conforms to professional standards of the scientific literature. Below I have made a few specific suggestions about wording, grammar, and clarity.

Line 62  Avoid the word “defective” because that’s not really the case. How about: “Since the current therapeutic medications have significant drawbacks...”

Line 64  “Many peptides have been demonstrated to have impressive antidiabetic effects.”

Line 69  Unclear what this means: “scattered in bioinformatics resources.”

Lines 70-71  “Fortunately, BioDADPep, a database dedicated to storage and management of antidiabetic peptides, have been proposed.” What does it mean that the database is proposed? Do you mean it has been developed? Can you say one more sentence about what it is and what it does?

Line 95  “...where 2025 peptides had between 5-50 amino acids.”

Line 127  “For a peptide sequence with L amino acids, if k = 0,...” (Small “i” in “if”)

Line 138  Unclear what this means: “There could be a part of key features which make a significant contribution....” Are you trying to say that the features needed to be cleaned up into order to ixdentify any significant contributions to the prediction or anti-diabetic peptides?

Line 143  “which were generated”

Lines 207-209  I suggest: “The RF/Pearson combination outperformed the other five combinations for all evaluation indicators except AUCROC.”

Lines 215-217  I suggest: “It is observed that the RF-based predictor outperformed the other three other machine learning predictors for all seven evaluation metrics. All seven evaluation metrics reached the highest level.”

Experimental design

Here I must insert the disclaimer that I am not a computational researcher, and I do not know a lot about computational or bioinformatics methodology. I hope that the other reviewer(s) was/were able to critically analyze the methods used in the study. For my part, I can say that the experimental design was clearly described and seemed to follow the conventions of the particular practices employed (at least based on the things I looked up while I was reading). It also appeared to me that the experiments were described in enough detail that they could be reproduced by another researcher who has a knowledge of computational methods. Additionally, the investigation appeared to be performed with high technical and ethical standards. In terms of identifying the knowledge gap that is filled by this contribution, the authors clearly stated that AntiDMPpred is the first tool capable of identifying antidiabetic peptides from sequence information.

Here are some suggestions to improve the reader’s understanding of the methods used to construct the datasets and extract the features:

Lines 88 + 94  Can you provide a little more information/description about both the BioDADPep and AVPdb databases?

Line 89  Can you say more about non-natural peptides versus natural peptides? How did they all come to be in the BioDADPep database? Why did you exclude the non-natural ones?

Line 131  Where does the number 25 come from?

Lines 132-133  This is too short to be a paragraph. Can you expand it with a little more description of the feature vector methodology? For example, where does the number 519 come from?

Validity of the findings

Again, I rely on the other reviewer(s) to evaluate the validity the computational findings, but I believe they were presented very clearly. I just have one question about the description of the results: in lines 218-219, why couldn’t the ROC curve be drawn or the AUCROC be calculated?

I did practice using the online tool, and I thought it was convenient and easy to use. I would suggest making the “Predict” button stand out more; perhaps it could be bright green. With all the buttons being gray, I felt I was naturally drawn to click on the “Browse” button simply because it is higher up, even though I was not uploading a file. I think a bright color that stands out for the “Predict” button would be very helpful.

Additional comments

This paper is a well-constructed and smoothly written, and by all accounts AntiDMPpred will be very useful web-tool. With the small corrections above, I think it will turn out to be an excellent publication.

---

## Round 0.2 · Minor Revisions

There are two minor suggestions made by reviewer 2, these can be readily addressed by some text changes in a revision.

Once these are completed I will accept the paper without further reviewer input.

Congratulations on an interesting and exciting study with a tool I am sure many will find useful.

·

Basic reporting

The points that I raised in the previous review have been resolved by the authors.

Experimental design

The points that I raised in the previous review have been resolved by the authors.

Validity of the findings

The points that I raised in the previous review have been resolved by the authors.

Additional comments

I thank the authors for their constructive, thorough and meaningful rebuttal and for their care in the revisions that have been made to the manuscript. The research has proceeded from an innovative idea to the implementation of a very useful resource, now reported in an excellent paper.

·

Basic reporting

The manuscript, AntiDMPpred: a web service for identifying anti-diabetic peptides, by X. Chen, B. He, and J. Huang, reports a new web-based tool that predicts anti-diabetic peptides. This tool has the potential to aid researchers in identifying lead peptides for further studies of bioactive molecules aimed at the treatment of diabetes.

The manuscript is assembled in a professional manner, with sections appropriate for the content of the study. It is self-contained, and the amount of content is appropriate for one publication unit. It is organized, generally well-written, and conforms to professional standards of the scientific literature.

I thank the authors for the corrections they made based on my first review, as well as that of the other reviewer. For the newly added text, I only have two very small suggestions about wording:

Line 331-331: Remove the word “receiving”: “There has been growing interest in using machine learning...”

Line 335: Add the word “an”: “However, an anti-diabetic predictor is currently lacking...”

Experimental design

Here I repeat the disclaimer that I am not a computational researcher, and I do not know a lot about computational or bioinformatics methodology. Clearly, the other reviewer was able to critically analyze the methods used in the study and make helpful suggestions that were addressed by the authors. From what I can surmise, the experimental design is clearly described and seemed to follow the conventions of the particular practices employed (at least based on the things I looked up while I was reading). It also appears me that the experiments are described in enough detail that they could be reproduced by another researcher who has a knowledge of computational methods. Additionally, the investigation appears to be performed with high technical and ethical standards. In terms of identifying the knowledge gap that is filled by this contribution, the authors have clearly stated that AntiDMPpred is the first tool capable of identifying antidiabetic peptides from sequence information.

The authors have addressed all the suggestions and concerns I put in my review. If the other reviewer is satisfied with how they addressed his suggestions, then I am also content.

Validity of the findings

Again, I rely on the other reviewer to evaluate the improvements the authors have made based on his review.

I thank the authors for adjusting the “Predict” button. I think the interface is appealing and easy to use.

Additional comments

My first review was generally positive with only small suggestions, and the authors have successfully addressed all my concerns thoroughly. Very nice work!

Here I have only included two very small wording suggestions, and I would not need to see the article again to be confident in its acceptance.

---

## Round 0.3 · accepted · Accept

Thanks for the corrections. We are good to go: congratulations!